

# Crown-of-thorns starfish spines secrete defence proteins

Adam K. Hillberg[1,2],  Meaghan K. Smith[1,2],  Blake S. Lausen[1,2],  Saowaros Suwansa-ard[1],  Ryan Johnston[1,2],  Shahida A. Mitu[1,2],  Leah E. MacDonald[2], Min Zhao[1,2],  Cherie A. Motti[3],  Tianfang Wang[1,2],  Abigail Elizur[1], Keisuke Nakashima[4],  Noriyuki Satoh[4] and  Scott F. Cummins[1,2]

[1] Centre for Bioinnovation, University of the Sunshine Coast, Maroochydore, QLD, Australia
[2] School of Science, Technology and Engineering, University of the Sunshine Coast, Maroochydore, QLD, Australia
[3] Australian Institute of Marine Science, Townsville, Australia
[4] Marine Genomics Unit, Okinawa Institute of Science and Technology Graduate University, Onna, Okinawa, Japan

## ABSTRACT

**Background**. The crown-of-thorns starfish (COTS; *Acanthaster species*) is a slow-moving corallivore protected by an extensive array of long, sharp toxic spines. Envenomation can result in nausea, numbness, vomiting, joint aches and sometimes paralysis. Small molecule saponins and the plancitoxin proteins have been implicated in COTS toxicity.

**Methods**. Brine shrimp lethality assays were used to confirm the secretion of spine toxin biomolecules. Histological analysis, followed by spine-derived proteomics helped to explain the source and identity of proteins, while quantitative RNA-sequencing and phylogeny confirmed target gene expression and relative conservation, respectively.

**Results**. We demonstrate the lethality of COTS spine secreted biomolecules on brine shrimp, including significant toxicity using aboral spine semi-purifications of >10 kDa ($p > 0.05$, 9.82 μg/ml), supporting the presence of secreted proteins as toxins. Ultrastructure observations of the COTS aboral spine showed the presence of pores that could facilitate the distribution of secreted proteins. Subsequent purification and mass spectrometry analysis of spine-derived proteins identified numerous secretory proteins, including plancitoxins, as well as those with relatively high gene expression in spines, including phospholipase A2, protease inhibitor 16-like protein, ependymin-related proteins and those uncharacterized. Some secretory proteins (*e.g.*, vitellogenin and deleted in malignant brain tumor protein 1) were not highly expressed in spine tissue, yet the spine may serve as a storage or release site. This study contributes to our understanding of the COTS through functional, ultrastructural and proteomic analysis of aboral spines.

# INTRODUCTION

The crown-of-thorns starfish (COTS; *Acanthaster species*), belongs to the Acanthasteridae family. They are efficient consumers of scleractinian coral (hard coral) species, and

Corresponding author
Scott F. Cummins,
scummins@usc.edu.au

when in outbreak proportions can cause extensive coral depletion resulting in significant directional shifts in coral reef communities (*Pratchett et al., 2014*). Located throughout the Indo-Pacific, COTS can grow up to 60–70 cm, with a dorsal surface covered in a dense, calcareous skeleton (*Birkeland, 1989*). COTS can release toxins into the water upon stress that cause respiratory complications in some marine organisms (*e.g.*, fish predators) (*Lee et al., 2013a*). Also, their array of spines, which are very sharp, are highly toxic upon touch. As a toxic and well-armoured animal, COTS have few natural predators (*Cowan et al., 2017*).

Five main types of spines have been characterised from the COTS body wall based primarily on size and shape: one aboral, one latero-oral and three that surround the oral (underside) surface (*Motokawa, 1986*). Spines that protect softer tissue, such as the tube feet and mouth, are typically more blunt and smaller than other spines. The sharp aboral COTS spine contains venom (*e.g.*, saponins) that has various bioactivities from hemolytic, hemorrhagic, capillary permeability, and increased hyperactivity of immune cells causing histamine release and mast cell degranulation (*Lee et al., 2013a*). To date, there has been one documented human casualty due to COTS venom poisoning; a female who died within 13 h post-envenomation from anaphylactic shock (*Ihama et al., 2014*). COTS venom is structurally unique compared to other characterised venoms and may, therefore, provide a new resource for novel drug identification. These toxins are released from the tissue (*Demeyer et al., 2015*; *Kitagawa & Kobayashi, 1978*), when a starfish is stressed and are capable of harming marine animals by inducing hemolytic effects together with engulfment-mediated DNA degradation (*Lee et al., 2013b*; *Ota et al., 2006*).

Similar to other starfish, COTS spine venom is known to contain saponins, which are amphipathic glycosides grouped as sapogenins that form a soapy lather upon contact with water (*De Marino et al., 1998*; *Ngoan et al., 2015*; *Thao et al., 2013*). COTS spine venom also contains plancitoxins, proteins of 20,000–25,000 Da (*Shiomi et al., 1988*), as well as various other small biomolecules that have recognised anti-cancer properties, including 2,2-azinobis-3-ethylbenzothiazoline-6-sulphonic acid, 1,1-diphenyl-2-picrylhydrazyl, $Fe^{2+}$ and butanol, all demonstrating inherit anti-melanoma properties (*Lee et al., 2014b*; *Shiomi et al., 1990*). Thus, COTS spine venom may provide a valuable source of anti-cancer compounds (*Lee et al., 2014a*).

Knowledge regarding COTS physiology and toxicology remains limited, which also extends to the biosynthesis and release mechanisms of their toxins. To date, one lethal factor (*Shiomi et al., 1988*), two class-I PLA2s (*Shiomi et al., 1998*) and one anticoagulant factor (*Karasudani et al., 1996*) have been purified from COTS spines and their properties have been clarified to some extent. In addition, COTS-conditioned water during conspecific aggregation and following stress (exposure to Giant triton snail), showed considerable differences in protein profiles (*Hall et al., 2017*). In this study, we investigated a lethal effect of COTS aboral spine protein preparations, indicating that protein toxins can be secreted into surrounding water. Subsequently, a proteotranscriptomic analysis has identified various excretory-secretory proteins (ESPs).

## MATERIALS AND METHODS

### Animals

COTS were collected under permit G17/38293.1 by the Association of Marine Park Tourism Operator divers working on the Great Barrier Reef, Cairns region, Northern Queensland. Starfish (∼25 cm in diameter) were transported by air freight to the University of the Sunshine Coast Aquarium facility (Sippy Downs), where they were housed for up to two weeks in a recirculating protein skimmed saltwater system of approximately 1,500 L capacity.

### COTS venom extraction for lethality bioassay

Two COTS venom extracts, 'Spine Extract' (SpE) and 'Stress Water Extract' (StWE), were prepared and their cytotoxicity tested. The SpE was prepared from 20 aboral spines collected from three different adult COTS ($n = 60$, unknown sex). Spines were combined and placed into 50 ml of Milli-Q water, then agitated for 1 h to facilitate venom release. A 14 ml aliquot was separated through a 10-kDa cut-off Amicon ultra-15 centrifugal filter unit (Merck, Australia) under centrifugation (4000 xg for 10 min) at 4 °C. Fractions which contained <10 and >10 kDa proteins (designated as <10-kDa SpE and >10-kDa SpE, respectively) were separately collected into fresh tubes and used immediately in a brine shrimp lethality bioassay. To obtain StWE, two adult COTS (unknown sex) were placed into individual glass tanks filled with 1 L of aerated filtered seawater (FSW). Animals were intermittently mechanically agitated by physical tapping with blunt forceps for 1 h. The conditioned seawater (2 × 1 L) was drained from each tank and strained through a 100-µm mesh to remove any particulate matter. The two samples were then subjected to a mobile phase filtration through a 0.45-µm membrane filter (Waters, Milford, MA, USA) under vacuum. Each filtered sample (32 ml; designated as StWE) was subsequently fractionated through a 10-kDa cut-off Amicon ultra-15 centrifugal filter unit (Merck, Rahway, NJ, USA) by centrifugation (4000 × g for 10 min) at 4 °C. After centrifugation, the solution retained on the membrane filter (>10-kDa StWE) and the eluted solution (<10-kDa StWE) were collected. Samples were immediately used in the brine shrimp lethality bioassay. All protein concentrations were quantified based on the absorbance at 280 nm using the Nanodrop 2000 instrument (Thermo Fisher Scientific, Waltham, MA, USA) after filtrations.

### Brine shrimp lethality bioassay

A brine shrimp lethality bioassay was performed according to *Meyer et al. (1982)*, with some modification. Brine shrimp eggs (*Artemia salina*) were hatched in a hatching container filled with 300 ml of aerated FSW for 48 h at room temperature. At 48 h post-hatch, brine shrimp larvae were transferred *via* a pipette into a 96-well plate filled with 20 µl/well of aerated FSW (N = 14 ± 3 individuals per well). Brine shrimp larvae were then exposed to the different extracts, added at various concentrations (5 replicates per dose per extract). For each extract, the higher concentrations used in the brine shrimp lethality assay were prepared at 1:2 of the original concentration of a given extract in each replicate (which was obtained from our preparation procedure described in the COTS venom extraction methods section). Two additional dilutions (1:10 and 1:100 of the upper concentration)

were also tested for LD50 calculations. For negative controls, FSW (20 µl) was added into the well.

    *Artemia* larvae were counted at 0, 24, and 48 h post-treatment to calculate percent mortality. Data was visualised by applying a log10 function to the COTS venom concentrations (*x*-axis) and plotting against the average shrimp mortality rate converted into probits (*y*-axis) (*Finney, 1952*). The lethal dose required to cause 50% mortality in *Artemia* (LC50) was calculated *via* a 3-point linear regression. Statistical analysis of data was performed using the IBM SPSS software. A one-way ANOVA (univariate analysis) and subsequent Scheffe *post-hoc* tests were used to establish significance at a confidence threshold of 95% ($P \leq 0.05$). Where data did not contain an equal variance of error according to Levene's test ($P \leq 0.05$), the Games-Howell *post-hoc* test was used to evaluate significant differences between different groups at a confidence threshold of 95% ($P \leq 0.05$).

## Histology and scanning electron microscopy (SEM)

At least four spines, removed from COTS using surgical scissors, were immediately placed into 4% paraformaldehyde and left overnight at 4 °C before being transferred and stored in 70% ethanol (EtOH). Spines were rehydrated in 50% EtOH (two washes; 30 min each) followed by autoclaved distilled water (three washes; 20 min each). Samples were decalcified by soaking in Morse's solution (10% sodium citrate and 50% formic acid) overnight before being washed with autoclaved distilled water to quench the decalcification reaction. Samples were then processed through increasing ethanol concentrations (70%–100%) and 100% xylene, before being embedded in paraffin wax. Embedded tissue blocks were sectioned at 8–12 µm-thick using a Leica microtome, and sections placed onto SuperFrost slides (Thermo Fisher Scientific, Waltham, MA, USA) and dried overnight at 37 °C. Hematoxylin and eosin (H&E) staining was completed following a routine protocol with modifications (*Smith et al., 2018*). Finally, sections were cleared using xylene solution, before being mounted with a coverslip using dibutylphthalate polystyrene xylene (DPX)-based mounting medium. Slides were viewed under a Leica DM550 microscope equipped with a Leica camera.

    For scanning electron microscopy (SEM), at least 4 aboral spines were removed using surgical scissors and immediately fixed with 10% formalin for overnight at 4 °C. After initial fixation, the spines were additionally fixed using a glutaraldehyde and paraformaldehyde buffer (pH 7.4) for 4 h. Spines were then cleaned 3 times (10 min each) using 0.2 M Millonig's buffer (pH 7.4) before and after immersion in osmium tetroxide buffer for 1 h. The spines were then dehydrated in a graded series of aqueous ethanol (EtOH; 50% to 100%), dried using a $CO_2$ critical point dryer, and platinum-coated using an Eiko IB-5 Sputter Coater onto an SEM mount. Four specimens were viewed using a JEOL 6300 field emission SEM at 15 kV.

## COTS spine protein extraction and preparation for mass spectrometry (MS)

To maximise the proteins identified from COTS spine, spines from both relaxed and stressed individuals were investigated to account for any potential differences in secretion.

Three adult COTS were held in isolation in aerated tanks (approximately 40 L in volume; one individual per tank) with artificial saltwater (salinity 35 ppt) to 30% capacity (12 L). Animals were acclimatised over a 6 h period with hourly 50% water changes using FSW. Acclimation was complete when surface foam and mucus were no longer visible (animals in this condition were considered 'relaxed'). Spines from each individual relaxed COTS ($n = 4$ spines per individual) were removed from the aboral side of the body using surgical scissors, placed in 10 ml of FSW, then gently mixed for 30 min. The same three adult COTS were stressed by mechanical agitation for 5 min, after which observable indicators of stress (including mucus production, formation of white foam on the water surface and increased movement), were observed. Spines ($n = 4$ spines per stressed individual) were then collected, immersed in 10 ml of FSW, then gently mixed for 30 min. Following mixing, the spine-conditioned FSW was acidified with trifluoroacetic acid (TFA) to a final concentration of 0.1%, then stored at $-80\,°C$ until preparation for mass spectrometry.

Acidified samples of both relaxed and stressed COTS spine-conditioned water were individually semi-purified using a Sep-Pak Vac 20cc (5 g) C18 cartridge (Waters, Milford, MA, USA), according to the manufacturer's instructions. Following sample loading and washes, samples were eluted using 5 ml 60% ACN/0.1% TFA. The eluent was dried in a speedvac (Savant, Barnstable, MA, USA) and stored at $-20\,°C$. The extracts were reconstituted in 200 µl aqueous 0.5% formic acid ($FA_{(aq)}$) in MilliQ water and an in-solution trypsin digest performed using trypsin solution (Promega, Madison, WI, USA), following the method described previously (*Ni et al., 2018*).

## Micro-high-pressure liquid chromatography-tandem quadrupole time-of-flight mass spectrometry (µHPLC qTOF-MS/MS) analyses and protein identification

The tryptic peptides were analysed as described by *Hall et al. (2017)*. In brief, µHPLC-MS/MS on a ExionLC liquid chromatography system (AB SCIEX, Concord, Canada) coupled to a qTOF X500R MS (AB SCIEX, Concord, Canada) equipped with an electrospray ion source. Each sample (20 µl) was injected onto a 100 mm × 1.7 µm Aeris PEPTIDE XB-C18 100 µHPLC column (Phenomenex, Sydney, Australia) equipped with a security guard column. Peptides were eluted using solvent A, 0.1% $FA_{(aq)}$, and solvent B, 100% ACN:0.1% $FA_{(aq)}$, at 400 µL/min flow rate. A linear gradient of 5–35% solvent B over 10 min was applied followed by a steeper gradient from 35% to 80% solvent B in 2 min and 80% to 95% solvent B in 1 min. Solvent B was held at 95% for 1 min to wash the column then returned to 5% solvent B for equilibration prior to the next sample injection. The ionspray voltage was set to 5500 V, declustering potential 100V, curtain gas flow 30, ion source gas 1 40, ion source gas 2 50 and spray temperature at 450 °C. MS data was acquired in the Information Dependant Acquisition mode. Full scan qTOF-MS data was acquired over the mass range 350–1400 *m/z*; product ion MS/MS data was acquired over 50–1800 *m/z*. Ions observed in the qTOF-MS scan exceeding a threshold of 100 cps and a charge state of +2 to +5 were set to trigger the acquisition of product ion. The data was acquired and processed using SCIEX OS software (AB SCIEX, Framinham, MA, USA). Biological triplicates were used for the analysis.

The µHPLC qTOF-MS/MS data were imported to the PEAKS studio (Bioinformatics Solutions Inc., Waterloo, ON, Canada, version 7.0) with the assistance of MS Data Converter (Beta 1.3, http://sciex.com/software-downloads-x2110). Peptides were analysed using PEAKS v7.0 (BSI, Canada) against the protein database assembled from genome data (http://marinegenomics.oist.jp) (*Hall et al., 2017*). *De novo* sequencing of peptides, database searches and characterisation of specific post-translational modifications were used to analyse the raw data; false discovery rate was set to $\leq$ 1%, and $[-10 * \log(p)]$ was calculated accordingly where p is the probability that an observed match is a random event. The PEAKS used the following parameters: (i) precursor ion mass tolerance, 0.1 Da; (ii) fragment ion mass tolerance, 0.1 Da (the error tolerance); (iii) tryptic enzyme specificity with two missed cleavages allowed; (iv) monoisotopic precursor mass and fragment ion mass; (v) a fixed modification of cysteine carbamidomethylation; and (vi) variable modifications including lysine acetylation, deamidation on asparagine and glutamine, oxidation of methionine and conversion of glutamic acid and glutamine to pyroglutamate.

## RNA-seq, gene expression and annotation

For initial relative expression analysis, transcriptome data for each of the tissues represented in http://marinegenomics.oist.jp/cots was downloaded and heatmaps constructed using R (version 3.1.1) (https://www.r-project.org/) based on Z-score values using the scale function. More in-depth RNA-seq gene expression quantification was performed using COTS radial nerve cord, tube foot and sensory tentacle derived from *Roberts et al. (2017)* and *Smith et al. (2017)*. In addition, COTS were randomly selected for aboral spine and stomach RNA-seq. Spines were cut three mm from the middle section along the aboral arm at the base of the spine, and then immediately transferred to RNAlater (Thermo Fisher Scientific, Waltham, MA, USA) before storing at −20 °C. Stomach tissue was obtained by inducing stomach eversion using small (<5 cm) coral fragments, when eversion was visually confirmed, ~200 mg of stomach tissue was cut and immediately put into cold RNAlater (Thermo Fisher Scientific, Waltham, MA, USA) before storage at −20 °C. Approximately 100 mg of the individual tissues from spine and stomach, was later used for RNA extraction using TRIzol™ Reagent (Thermo Fisher Scientific, Waltham, MA, USA), as per manufacturers protocol, then total RNA was quantified using a Nanodrop 2000 (Thermo Fisher Scientific, Waltham, MA, USA) and integrity was assessed with a Bioanalyzer RNA 6000 Nano mRNA kit (Agilent Technologies, Santa Clara, CA, USA). Only samples with a RIN value >4 were deemed suitable for RNA-seq. Total RNA from aboral spine ($n = 6$) and stomach ($n = 3$) was sent to Novogene (Hong Kong) for Illumina 2500 sequencing using their standard workflow (https://en.novogene.com/).

Full-length precursor sequences of proteins identified in the MS analysis of spine samples were retrieved from the protein database assembled from the available COTS transcriptomes/genome data, and subsequently used for the prediction of putative secreted proteins. Putative secreted proteins were predicted based on the presence of signal peptide and absence of transmembrane domain in the protein precursor sequences, using SignalP (V.5.0) and TMHMM (V.2.0) webtools (*Almagro Armenteros et al., 2019*; *Krogh et al., 2001*). Tissue expression of genes encoding identified putative secreted proteins was

investigated using the publicly available transcriptomes from the COTS genome browser (http://marinegenomics.oist.jp) (*Hall et al., 2017*). This comprised of COTS tube foot, mouth, spine, sensory tentacle, radial nerve cords (male and female), testis, ovary, egg, mid-gastrula tissues (*Hall et al., 2017*; *Roberts et al., 2017*). Gene expression in TPM (transcripts per kilobase million) values was transformed to the z-scores before being visualized in a heat map using the ClustVis webtool (*Metsalu & Vilo, 2015*).

General functions of identified putative proteins were investigated based on the Gene Ontology (GO) annotation. In brief, full-length sequences of putative secreted proteins were annotated against a non-redundant database on the National Center for Biotechnology Information (NCBI) using a BLASTp search on the Omicsbox software (BioBam). GO annotation, limited to the biological process category, was then performed. The top-40 most abundant GO terms were displayed in a WordCloud format where the front size represented the node score calculated according to the following formula: $\text{score} = \sum_{GOs} \text{seq} \times \alpha^{dist}$, where 'seq' is the number of different sequences annotated at a child GO term and 'dist' the distance to the node of the child (BioBam).

Based on GO and literature, plancitoxin, phospholipase A2 and peptidase inhibitor 16-like were recognised as toxins. Relative expression of plancitoxin, phospholipase A2 and peptidase inhibitor 16-like genes in tissues was determined using the CLC Genome Workbench (version 20) based on transcripts per kilobase million (TPM), utilizing the COTS genome Version 1.0 (http://marinegenomics.oist.jp/cots), as reference. The statistical power of this experimental design, calculated in RNASeqPower was 0.725.

### Phylogenetic analysis

Echinoderm non-COTS sequences used for phylogenetic analysis were curated from the NCBI protein database. Deduced protein sequences were aligned in MEGA v.7 and v.11 using the CLUSTAL algorithm and phylogenetic trees constructed using the maximum likelihood method with 1,000 bootstrap replicates. iTOL online tool (*Letunic & Bork, 2021*) was used to assist in phylogenetic tree illustration.

## RESULTS

### Toxicity of COTS venom extracts

COTS aboral spines ($\sim$3 cm in length; Fig. 1A) were collected and processed for venom collection (SpE), in addition to stress water extract (StWE). Brine shrimps were exposed to varying concentrations of crude and semi-purified extracts, then activity was observed at 48 h post-exposure (Fig. 1B). Brine shrimp exposed to FSW (as negative control) showed a mortality of 3.48 $\pm$ 3.28% (mean $\pm$ SD). Crude SpE at 110 µg/mL induced significantly higher mortality (52.93 $\pm$ 14.63%) when compared with the control group, however, dilutions of 1:10 and 1:100 did not. Fractionation of SpEs (<10 kDa and >10 kDa), demonstrated that only >10 kDa at 93.3 µg/ml contained significant toxicity (51.33 $\pm$ 13.13%), comparable to that of crude SpE. Crude StWE induced significant mortality at 107 and 10.7 µg/ml (88.21 $\pm$ 9.64% and 50.08 $\pm$ 11.45%, respectively), while fractionated StWE at >10 kDa caused significant mortality (81.52 $\pm$ 9.81%) at 98.2 µg/ml. Although <10 kDa StWE did induce mortality at the highest concentration tested (10.0 µg/ml), the data

**Table 1  A summary of the venom protein extracts prepared from the crown-of-thorns starfish (COTS) and their cytotoxicity based on a brine shrimp lethality bioassay.** Protein concentration was measured in μg/ml. See Table S1 for raw data. See Table S1 for raw data.

| Extract | LC50 (μg/ml) | Toxicity class (Meyer's/Clarkson's) |
|---|---|---|
| Spine extract (SpE) | 153.65 | Toxic/Medium toxic |
| <10 kDa SpE | 1.43E+23 | Non-toxic |
| >10 kDa SpE | 187 | Toxic/Medium toxic |
| Stress-conditioned seawater extract (SWE) | 14.42 | Toxic/Highly toxic |
| <10 kDa SWE | 12.98 | Toxic/Highly toxic |
| >10 kDa SWE | 30.74 | Toxic/Highly toxic |

showed an unequal variance of error according to Levene's test (Games-Howell *post-hoc* test was considered insignificantly different to the control group ($P = 0.063$)). The LC50 of each extract and the extract toxicity are summarized in Table 1. All extracts, except the <10 kDa SpE, were considered toxic based on Meyer's criterion. Following Clarkson's criterion, the SpE and >10 kDa SpE exhibited medium toxicity, whereas all StWE treatments were considered highly toxic.

## Histology of COTS spines and proteomic analyses of spine-conditioned water

SEM images of COTS aboral spines provided a view of the basic ultrastructure of the outer surface of COTS spines with progressive magnifications (Figs. 2A and 2B), revealing a topography of craters and pores (2–10 μM in diameter) covering the entirety of the spine. A SEM image of a longitudinal sectioned spine revealed the underlying components, including an inner core consisting of a porous three-dimensional meshwork of endoskeleton plates (trabeculae), which aligned longitudinally and parallel to the spine's length, and an outer epithelial layer (Fig. 2C). This was similarly observed through histological section analysis (Fig. 2D), and further demonstrating an outer columnar epithelium with a thick basement membrane. A well-defined nucleus was present and granular-like dots could be observed, dispersed throughout the basement membrane.

Aboral spine proteins were extracted from combined relaxed and stressed COTS individuals to ensure the full repertoire of potential toxin proteins were identified following mass spectrometry. In total, 157 protein sequences were identified that contained at least one MS peptide match (Table S2), from which there were 56 different proteins. GO annotation indicated that identified proteins were mainly associated with cellular process, lipid transport, cytoskeleton organisation and modulation of process of another organism (*e.g.*, induces hemolysis in another organism) (Fig. 3A). Sixteen proteins were consistently identified within the majority of spine extracts (both relaxed and stressed COTS spines) and predicted to encode a signal peptide (Fig. 3B, Table 2). Besides oki.111.24 which was annotated as an uncharacterised protein, other secreted proteins had high confidence matches to known proteins, including for example, ependymin-related proteins, plancitoxin-1, phospholipase A2, cysteine-rich secretory protein, pancreatic triacylglycerol lipase-like protein and vitellogenins. A cluster analysis based on gene expression pattern,

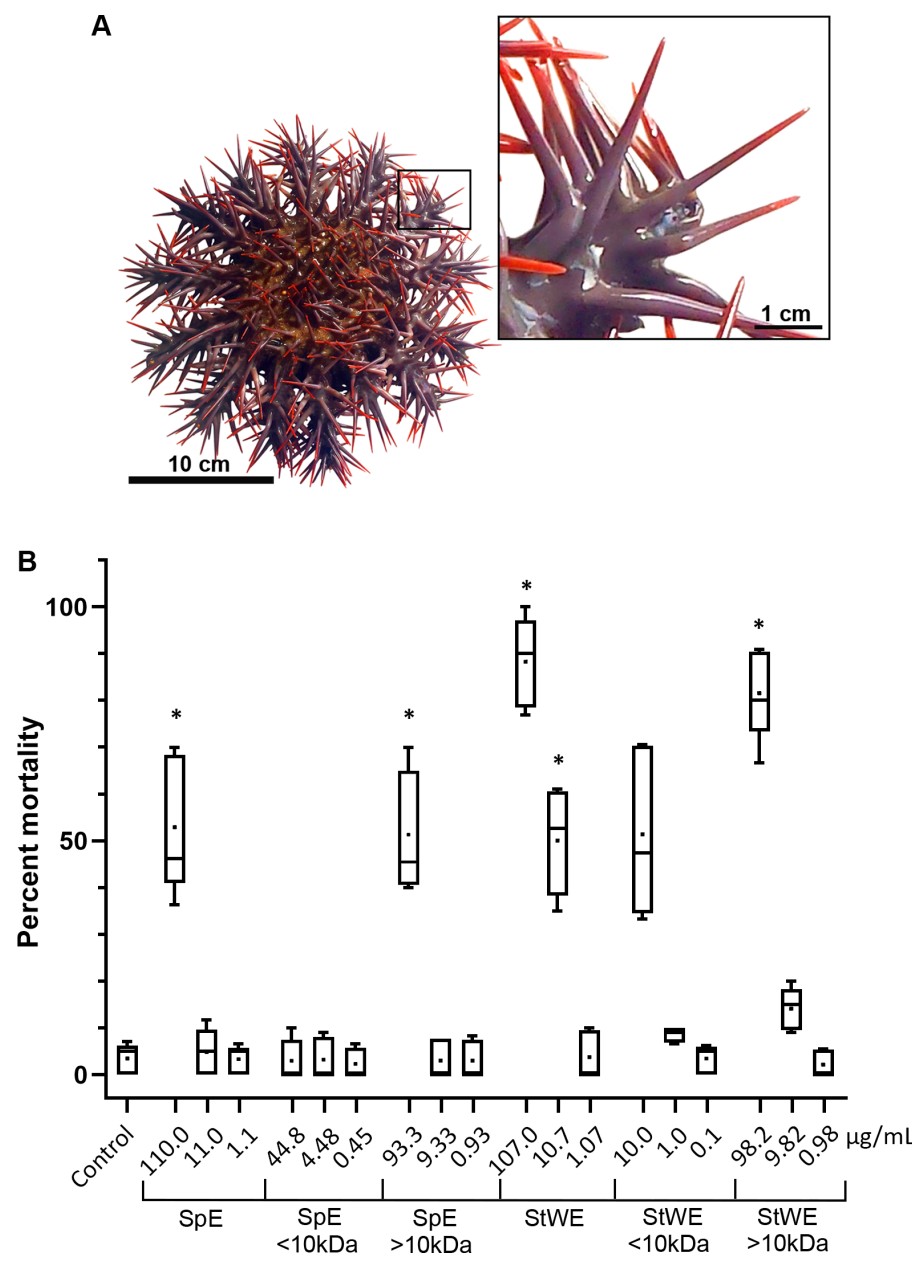

**Figure 1** **Toxicity activity of crown-of-thorns starfish (COTS) venom extracts based on a brine shrimp (*Artemia salina*) lethality bioassay.** (A) Anatomical photo of COTS showing aboral spines. Image in the inset shows a morphology of spines at higher magnification. (B) Percent mortality was determined 48 h post-exposure to COTS extract protein concentrations (μg/mL), and compared to controls (filtered seawater; FSW). 5 biological replicates/treatment. Error bars show standard deviation (SD) of data. Asterisks indicate significant difference to the control group ($p \leq 0.05$). SpE, spine extract; StWE, stress water extract.

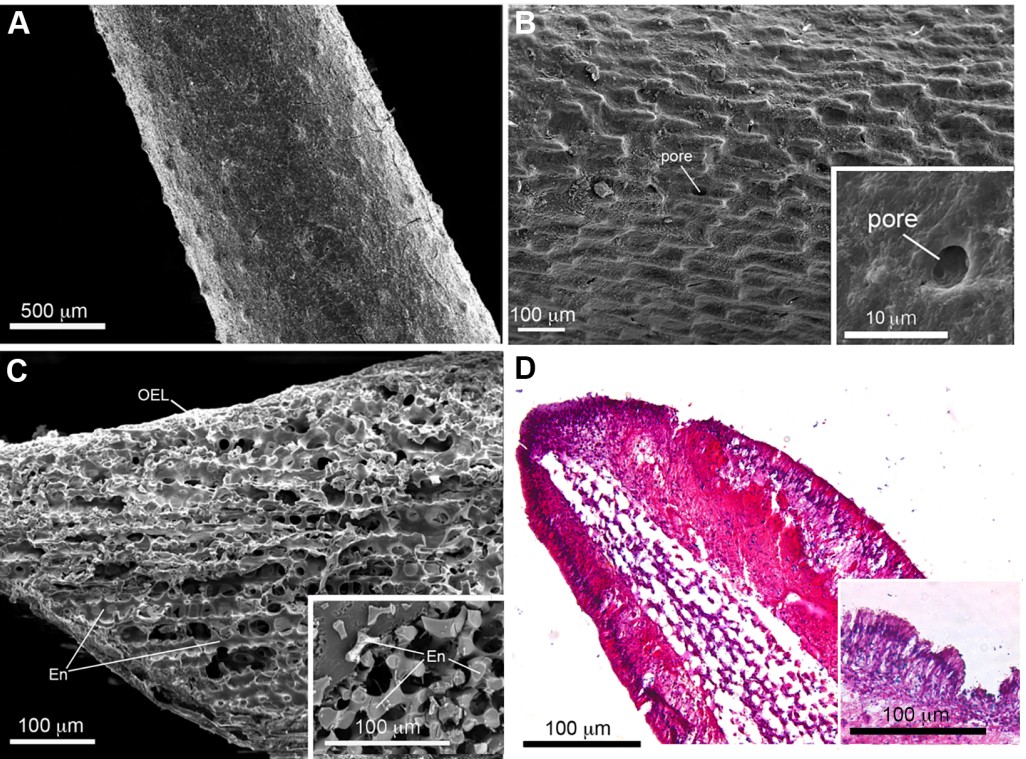

**Figure 2   Scanning electron microscopy (SEM) and histological analysis of crown-of-thorns starfish (COTS) aboral spine.** (A) SEM images showing COTS aboral spine outer surface, and at (B) higher magnification. Pores located on the spine surface are indicated. (C) SEM image of longitudinally sectioned aboral spine showing the core structure of spine, including endoskeletons (En) and outer epithelial layer (OEL). (D) Hematoxylin and eosin (H&E) stain of COTS aboral spine longitudinal section.

showed two distinct groups of genes, dependent on whether highly expressed in the spine (Fig. 3B), which were further investigated below.

## COTS spine excretory-secretory proteins
### *Plancitoxins*

The COTS genome models predict three plancitoxin genes, two with full-length (*oki.27.33* and *oki.27.35*) and one with partial length (*oki.27.34*), that encode proteins with deoxyribonuclease (DNase) II domain organisation (Fig. 4A). The deduced full-length forms were 356 amino acids in length which include the signal peptide sequences. Oki.27.33 shared highest similarity to oki.27.35 and to the previously identified COTS plancitoxin I (84.12%; GenBank accession number: BAD13432), whereas oki.27.34 shared ~36–38% similarity to oki.27.33 and oki.27.35. A DNase II domain (domain ID: IPR004947), catalytic domain repeats 1 and 2 (CD09120 and CD09121), and seven conserved cysteine residues were recognised in the full-length COTS plancitoxins (Fig. 4A). Gene expression of all forms were relatively abundant in the spine, although less, plancitoxins are also detected in other adult COTS tissues such as tube foot, sensory tentacle, RNC and stomach (Fig. 4B). Within the sensory tentacle, expression of *oki.27.34* was notably higher than other
**A**

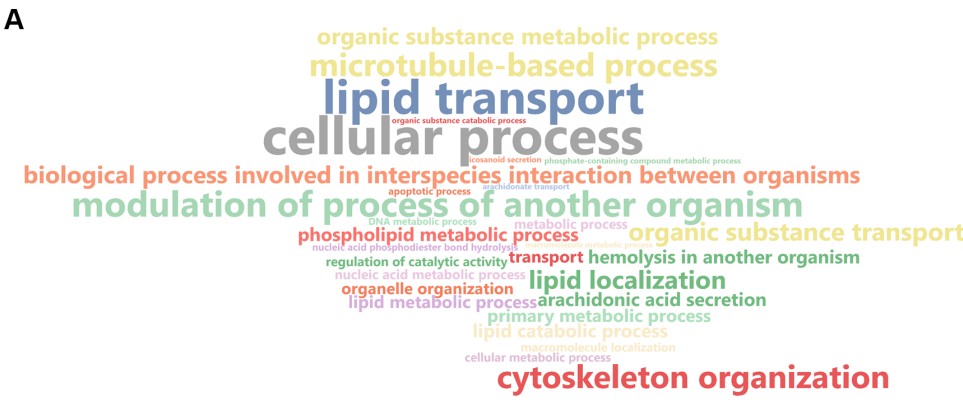

**B**

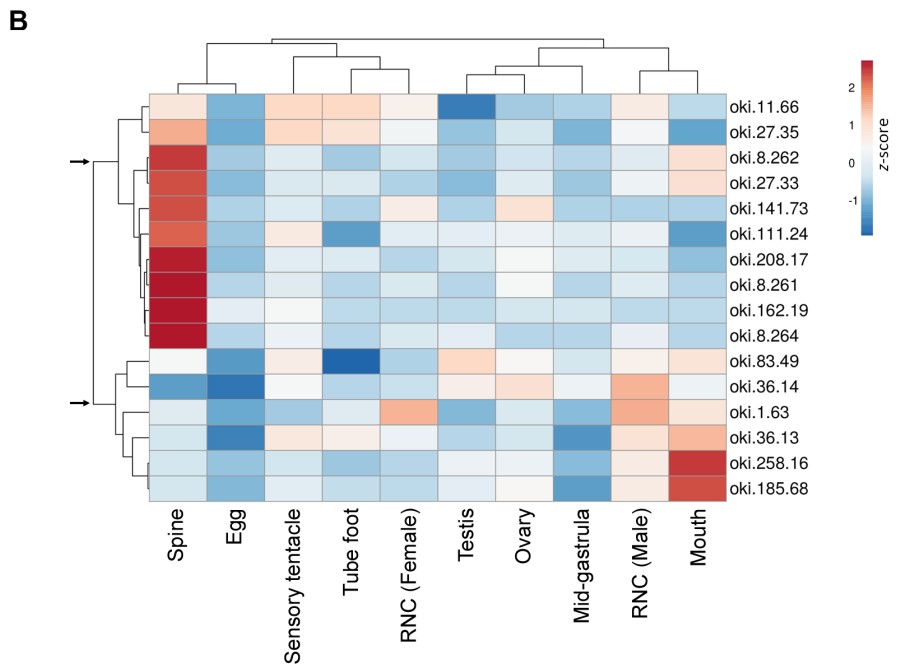

**Figure 3** **General biological functions of total proteins and gene expression of putative secreted proteins identified in the proteomic analysis of crown-of-thorns starfish (COTS) spines.** (A) WordCloud displaying the top 40 most abundant Gene Ontology (GO) terms (limited to the biological process category) for total proteins identified in the COTS spine extracts. The front size indicates the node score of each GO term. (B) Heatmap showing relative expression of genes encoding putative secreted proteins in various tissues. Two distinct clusters of genes are observed (arrows). Gene description of each gene ID is provided in Table 2.

forms. Only *oki.27.33* and *oki.27.35* were identified from aboral spine protein preparations. Phylogenetic analysis showed that COTS plancitoxins share similarity with plancitoxins reported in other echinoderms and molluscs, yet formed a distinct clade from vertebrate DNase II-alpha proteins and their homologous proteins in nematodes (Fig. 4C).

### Phospholipase A2
The COTS genome contained 8 genes encoding secreted-type PLA2 (sPLA2) proteins: *oki.8.260*, *oki.8.261*, *oki.8.262*, *oki.8.264*, *oki.8.268*, *oki.8.269*, *oki.8.270*, and *oki.36.73*. All

**Table 2  Summary of putative secreted proteins identified in the spines of COTS.** The numbers of total and unique mass spectral (MS) peptide match, BLAST hit proteins with corresponding species, e-value and GenBank accession numbers, and identified protein domains for each protein precursor are included.

| Protein ID | Number of total MS peptide matches | Number of unique MS peptide matches | BLAST hit protein and species (NCBI accession number) | e-value of BLAST hit | Identified protein domains |
|---|---|---|---|---|---|
| oki.11.66 | 1 | 1 | Ependymin-related protein 1-like [*Patiria miniata*] (XP038069568) | 1E−71 | Signal, Epdr domain |
| oki.27.35 | 23 | 17 | Plancitoxin-1-like [*Acanthaster planci*] (XP022085063) | 0 | Signal, Dnase II domains |
| oki.8.262 | 7 | 7 | Phospholipase A2-I-like [Acanthaster planci] (XP022079272) | 8E−102 | Signal, PA2c domain |
| oki.27.33 | 7 | 1 | Plancitoxin-1-like [*Acanthaster planci*] (XP022085064) | 0 | Signal, Dnase II domains |
| oki.141.73 | 1 | 1 | Ependymin [*Patiria miniata*] (XP038059830) | 2E−15 | Signal, Epdr domain |
| oki.111.24 | 2 | 2 | Uncharacterized protein LOC110983903 [*Acanthaster planci*] (XP022099251) | 0 | Signal |
| oki.208.17 | 1 | 1 | Cysteine-rich secretory protein LCCL domain-containing 2-like [*Acanthaster planci*] (XP0022106668) | 9.94E−133 | Signal, SCP domain |
| oki.8.261 | 5 | 1 | Phospholipase A2-II-like [*Acanthaster planci*] (XP022079273) | 7E−103 | Signal, PA2c domain |
| oki.162.19 | 3 | 3 | Pancreatic lipase-related protein 2-like [*Acanthaster planci*] | 3.23E−158 | Signal, Pancreatic lipase-like domain |
| oki.8.264 | 11 | 3 | Phospholipase A2-II-like [*Acanthaster planci*] (XP022079277) | 1E−102 | Signal, PA2c domain |
| oki.83.49 | 3 | 2 | Deleted in malignant brain tumors 1 protein-like [*Lytechinus variegatus*] (XP041455825) | 0 | Signal, SR, FTP, EGF domains |
| oki.258.16 | 26 | 26 | Kielin/chordin-like protein [*Patiria miniata*] (XP038067966) | 0 | Signal, von Willebrand factor type C domain |
| oki.36.13 | 6 | 4 | Vitellogenin 2 [*Patiriella regularis*] (AHK12748) | 0 | Signal, d1gw5a, LPD N, DUF1943, VWD domains |
| oki.36.14 | 2 | 2 | Vitellogenin 2 [*Patiriella regularis*] (XP022087200) | 0 | Signal, d1gw5a, LPD N, DUF1943, VWD domains |
| oki.185.68 | 56 | 5 | Vitellogenin-like [*Acanthaster planci*] (XP022105422) | 0 | Signal, d1gw5a, LPD N, DUF1943, VWD domains |
| oki.1.63 | 2 | 2 | Pentraxin fusion protein-like [*Acanthaster planci*] (XP_022090826) | 0 | Signal, F5/8 type C domain |

genes except *oki.36.73* were located on scaffold 8. Deduced proteins from sPLA2 genes on scaffold 8 shared ~50–58% similarity, whereas oki.36.73 showed highest similarity (41.29%) to oki.8.268. All COTS sPLA2 precursors (159–175 amino acids in length) displayed a signal peptide sequence, a PLA2 domain (domain ID: IPR016090), a catalytic PLA2 histidine active site (PROSITE patterns ID: PS00118), and 14 conserved cysteine residues (Fig. 5A). The deduced PLA2 from *oki.8.261* and *oki.8.262* shared 98.73% and 100% identity, respectively, to the previously reported COTS PLA2-I and -II (GenBank:

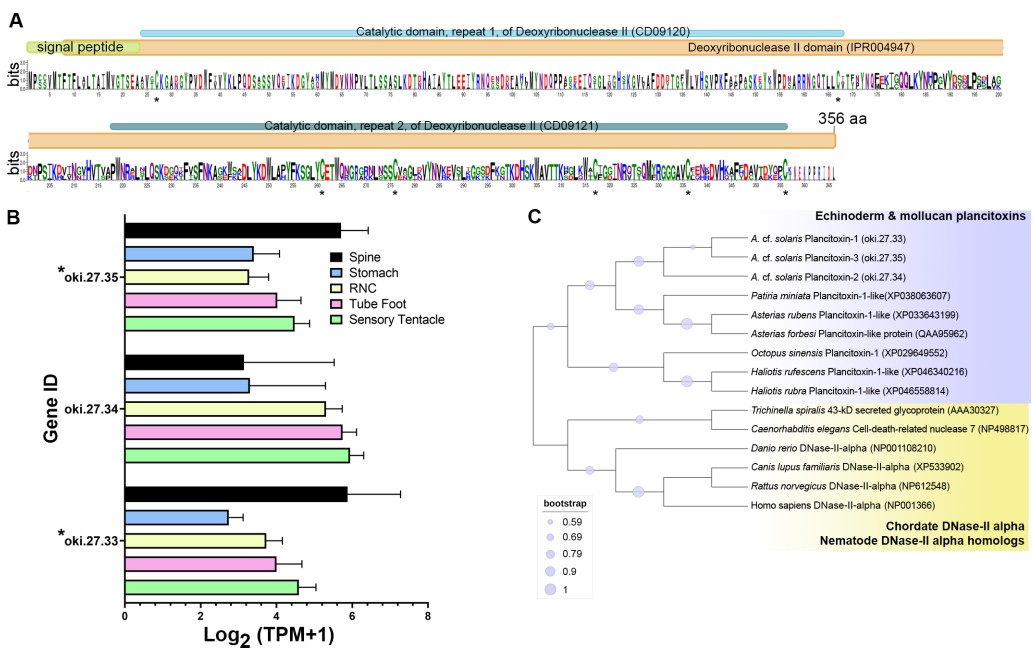

**Figure 4** **Plancitoxins in the crown-of-thorns starfish (COTS).** (A) Schematic representation of deduced COTS plancitoxins (oki.27.33, oki.27.34, and oki.27.35). Sequence logo shows the conservation of amino acid sequences among all forms of COTS plancitoxins. Coloured bars at the top of sequence logo display the conserved domains and signal peptide region, and black asterisks show the position of conserved cysteine residues. (B) Graph showing relative plancitoxin gene expression (transcripts per million, TPM) in COTS tissues. An asterisk (*) indicates significantly higher ($P < 0.05$) gene expression in spine compared to other tissues. Red asterisks denote those identified in this study by aboral spine protein analysis. (C) Phylogenetic tree of COTS (*Acanthaster* cf. *solaris*) plancitoxin proteins with plancitoxin/plancitoxin-like proteins, deoxyribonuclease-2-alpha (DNase-II-alpha) and DNase II-alpha homologs from other species (model, Whelan and Goldman [WAG]; 1,000 bootstraps). Purple circles on the clades indicate level of bootstrap confidence (≥50%).

BAE46765 and BAE46766) (*Ota et al., 2006*). Relative gene expression of COTS sPLA2 indicated that the *oki.8.260*, *oki.8.261*, *oki.8.262*, and *oki.8.264* were abundant in the spine, whereas *oki.8.268* was predominant in the sensory tentacle, tube foot, and radial nerve cords (both sexes), but also detected in the spine tissue (Fig. 5B). However, gene expression of *oki.8.269*, *oki.8.270*, were very low or not detected in any tissues investigated, besides *oki.36.73*, which showed some expression in stomach tissue. Among all sPLA2 identified, only protein products from *oki.8.261*, *oki.8.262*, and *oki.8.264* were detected from COTS aboral spine sample preparations (Fig. 3). Phylogenetic tree analysis of COTS sPLA2 proteins with different groups/types of sPLA2 in other species showed that echinoderm sPLA2 (including those of COTS) formed a distinct cluster, which was then rooted with PLA2 groups I (*e.g.*, type IA PLA2 toxin in a cobra and type IB PLA2 in humans) and IX (*i.e.*, PLA2 venom in a marine snail, *Conus magus*) (Fig. 5C). However, oki.36.73, which formed a separate root from other proteins, appeared to be distinct from other secreted PLA2 proteins. Another distinct clade (including humans) contained a group of sPLA2 proteins from various species which had been classified as PLA2 groups II, III, V and X.

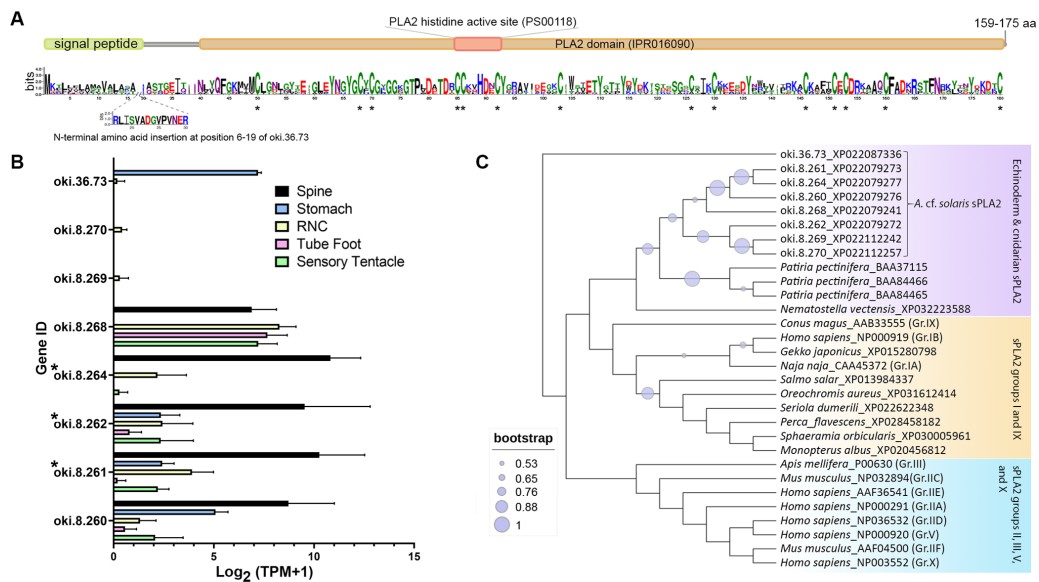

**Figure 5** Secreted phospholipase A2 (sPLA2) proteins in the crown-of-thorns starfish (COTS). (A) Schematic representation of deduced sPLA2 proteins identified in the COTS genome (gene IDs included). Sequence logo shows an amino acid sequence conservation among all forms. Coloured bars above a sequence logo shows the conserved domains and signal peptide region. An N-terminal insertion for oki.36.73 is shown from positions 6 to 19. Asterisks indicate highly conserved cysteine residues among COTS sPLA2, as well as sPLA2 in other animals. (B) Histogram showing relative COTS sPLA2 gene expression in TPM (transcripts per million) in COTS tissues. An asterisk (*) indicates significantly higher ($P < 0.05$) gene expression in spine compared to other tissues. Red asterisks denote those identified in this study by spine protein analysis. (C) Phylogenetic tree of COTS (*Acanthaster* cf. *solaris*) and other sPLA2 proteins. sPLA2 from different groups/types are included. The tree was constructed based on maximum likelihood method (model, WAG; frequency =5; 1,000 bootstraps). Purple circles on the clades indicate level of bootstrap confidence ($\geq$50%).

### Peptidase inhibitor 16-like

Oki.208.17 was annotated as a peptidase inhibitor 16-like protein (PI16) (Table 2), sharing greatest similarity to a PI16-like protein from *Patiria miniata* (accession number: XP038075979; 63.32% identity) and a PI16 found in in *Asterias rubens* (accession number: XP033625744; 52.36% identity). The oki.208.17 precursor was 220 amino acids in length and contained a signal peptide, cysteine-rich secretory protein (CRISP) domain (IPR001283), a venom allergen 5-like domain (IPR002413) and an allergen V5/testis-specific protein (Tpx-1)-related conserved site (IPR018244), corresponding to the '$D_{142}$HYTQLVWAKS$_{152}$' motif (Fig. 6A). Ten cysteine residues were present, eight of which were within the CRISP domain. Expression of *oki.208.17* was high in the spine tissue (almost 700 TPM), whereas low expression (<10 TPM) was also detected in other tissues (Fig. 6B). Phylogenetic tree analysis indicated that COTS oki.208.17 was most closely related to echinoderm PI16s, which formed a sister clade with vertebrate PI16/PI16-like proteins (Fig. 6C). Snake venom-type CRISPs and insect venom allergen 5, were more distantly related.

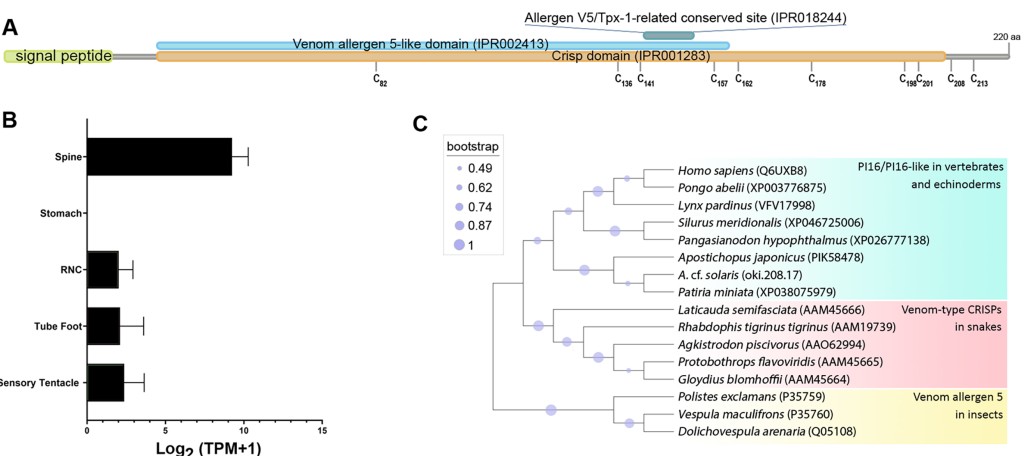

**Figure 6** **A peptidase inhibitor 16-like protein (PI16) in the crown-of-thorns starfish (COTS).** (A) Schematic representation of COTS PI16 (oki.208.17). Coloured bars show the conserved domains and signal peptide region. Cysteine residues ('C') and their amino acid positions are indicated. (B) Histogram showing relative gene expression (transcripts per million; TPM) of *oki.208.17* in COTS tissues. An asterisk (*) indicates significantly higher ($P < 0.05$) gene expression in spine compared to other tissues. (C) Phylogenetic tree of PI16s and CRISPs. Tree was constructed based on maximum likelihood method (model, WAG; frequency =5; 1,000 bootstraps). Purple circles on the clades indicate level of bootstrap confidence ($\geq$45%).

## Other excretory-secretory proteins identified from COTS aboral spine extracts

Multiple ependymin-related proteins (EPDRs) were identified from COTS spine sample preparations (Table 2; oki.11.66 and oki.141.73) were detected from spine extracts, and with their gene expression highest in the spines (Fig. 3B). Similarly, a pancreatic lipase-related protein (oki.162.19) and an uncharacterized protein (oki.111.24) are highly expressed in spines. The uncharacterized protein did not match to any other species in the NCBR nr database or had any recognised domains, however, the predicted mature protein was rich in cysteines (18 cysteine residues). Although identified from spine extracts, single DMBT1-like (oki.83.49) and kielin/chordin-like proteins (oki.258.16), as well as three Vtg-like proteins (oki.36.13, oki.36.14, and oki.185.68) that contained a 'vitellogenin_N superfamily' and 'VWD superfamily' motifs, were most highly expressed in the mouth and sensory tentacles, while spine expression was low (Fig. 3B). The secreted DMBT1-like protein contained 3 domains (Table 2): the F5/8 Type C domain (also known as discoidin domain), a calcium-binding EGF-like domain, and a scavenger receptor Cys-rich (SRCR) domain, which were repeated throughout the entire protein.

## DISCUSSION

Besides the calcareous endoskeleton-rich body wall that provides a first line of protection to most echinoderms, many echinoderms also bear spines and pincer-like pedicellariae which provide them additional protection against predators (*Lawrence, 1987*; *Pechenik, 2010*). COTS is an excellent example of a spiny echinoderm, since their bodies are heavily

covered with numerous sharp, long, and rigid spines. More importantly, COTS spines are also equipped with toxins which, having penetrated the tissue (*i.e.,* a puncture or sting), can cause cellular damage such as haemolysis and haemorrhaging (*Lee et al., 2013a*). While a small number of toxins have been identified and reported in COTS spines (*Ota et al., 2006*; *Shiomi et al., 2004*; *Shiomi et al., 1998*), no detailed proteomic analysis had been performed. Here, we initially investigated the lethality of biomolecules housed in COTS spines and in seawater following stress, followed by molecular proteomics of spines.

Brine shrimp lethality bioassays with crude spine extract supported the presence of potent and active toxins in the spines, as demonstrated previously (*Lee et al., 2013b*; *Sato et al., 2008*), and further showed that toxins >10 kDa could be secreted into the surrounding water that cause significant mortality. Since extracts >10 kDa were toxic, we inferred that the toxin(s) were proteinaceous. Based on previous studies, the protein toxins were likely to be, or are related to, the plancitoxins and PLA2 (*Ota et al., 2006*; *Shiomi et al., 2004*), as well as potentially novel toxins. Toxins present at <10 kDa may comprise asteroid saponins (or steroid glycosides) (*De Marino et al., 1998*; *Ngoan et al., 2015*; *Thao et al., 2013*), however, besides body wall components, these have not been characterised at the molecular level in the surrounding water. The toxic nature of seawater conditioned by stressed COTS, suggests they deliberately release protein toxins into the water, presumably as a chemical defensive mechanism to warn off predators, while the toxins on the spines provide protection against physical attack.

The histology of the COTS primary aboral and oral spines have has been described (*Motokawa, 1986*). Within the connective tissue located at the base of the aboral spine, three different cells are present, including neurosecretory cells that contain electron dense granules, nerve cells with granules, and cells with unidentified function (*Motokawa, 1986*). We used SEM and histological microscopy to describe the same COTS aboral spines that were subsequently used for proteomics analysis. We further described a three-dimensional meshwork of trabeculae (called stereom (*Smith, 1980*)) that showed a longitudinal arrangement parallel to the length of the spine, but with no apparent pore alignment, indicating the COTS spine is a galleried-type microstructure, imparting strength and rigidity to the spine. RNA-seq information obtained from the spines and used in this study, indicates that the spines are highly transcriptionally active. Based on the internal microstructure, we suggest that it is the external single layer of tall epithelial cells that produce the majority of transcripts, including the dense collagen fibres making up the connective tissue that were clearly observed underlying the epithelium.

Our proteomic analysis of COTS spines provided an insight into the proteins produced and/or released by the spines. A total of 56 different proteins were identified as basic structural proteins (actin, tubulin, and histone *etc.*) and enzymes (*e.g.,* metalloproteinase, helicase, carboxypeptidase D-like, serine/threonine-protein kinase), which also included the recognised toxins, plancitoxin and PLA2 (*Shiomi et al., 1998*). These were present within a distinct group of 10 secretory proteins most highly expressed in the spine tissue (see Fig. 3), which also included two pancreatic triacylglycerol lipase-like proteins, a PI16, two EPDRs and an uncharacterized protein (oki.111.24). Only the uncharacterized protein was not identified in any other animal species based on BLAST searches, strongly

suggesting it could play a role in species-specific conspecific communication. EPDRs have been described in COTS as potentially playing a role in conspecific communication due to their presence within the conditioned seawater of aggregating and alarmed COTS (*McDougall et al., 2018*). However, accumulating evidence based on mammalian EPDR structure, such as human EPDR1, has suggested that it interacts with anionic lipids of the cell membrane and may play a role in the breakdown/transport of anionic lipids (*Wei et al., 2019*). Anionic lipids have been associated with the cytotoxic properties of snake venom toxins (*Konshina et al., 2010*). Hence, we hypothesize that COTS EPDRs identified in the spines (oki.11.66 and oki.141.73) may contribute to the properties of COTS spine toxins through binding of small biomolecules.

In our study, two plancitoxins (1 and 3) were identified in spine protein preparations. COTS plancitoxins have previously been identified as enzymes in adult COTS related to the mammalian deoxyribonuclease II(DNase II) (*Shiomi et al., 2004*). COTS plancitoxin 1 was the first example of a toxin DNase that had the ability to cause severe damage to mice liver *via* caspase-independent apoptosis, as well as observable swelling of the gall bladder and necrosis of hepatocytes (*Lee et al., 2014a*). Previous studies showed that the crude toxin extracted from the COTS spines exhibits diverse biological activities, including lethality, hemolytic activity, myonecrotic activity, haemorrhagic activity, capillary permeability-increasing activity, edema-forming activity, PLA2 activity (*Shiomi et al., 1985*), histamine-releasing activity from mast cells (*Shiomi et al., 1989*), cardio-vascular actions (*Shiroma et al., 1994*; *Yara et al., 1992*) and anticoagulant activity (*Karasudani et al., 1996*). Taken together, there is little doubt that the plancitoxins identified in this study are part of the COTS toxin repertoire. Furthermore, the evolutionary analysis of plancitoxins with other DNase-II proteins/homologs using a phylogenetic tree analysis suggests that echinoderm and aquatic molluscan plancitoxins might have diverged from an ancestral DNase-II proteins/homolog, with further divergence of plancitoxins in aquatic molluscs and echinoderms having occurred thereafter. Currently, there is no evidence that molluscan plancitoxins (*e.g.*, abalone and octopus) act as toxins.

Three sPLA2's were identified from our protein preparations, however, several other genes encoding sPLA2 are distributed within close proximity in the genome. Phospholipases are a ubiquitous group of enzymes that hydrolyse glycerophospholipids (*Boyanovsky & Webb, 2009*). The sPLA2 acts at the sn-2 site of glycerophospholipids, resulting in fatty acid release and synthesis of lysophospholipids that are involved in G protein-coupled lipid signalling. The sPLA2 family provides a seemingly endless array of potential biological functions that is only now beginning to be appreciated. For instance, in humans, this family comprises 9 different members that vary in their tissue distribution, hydrolytic activity, and phospholipid substrate specificity (*Boyanovsky & Webb, 2009*). Ten members of the sPLA2 family have been identified in mammals, which are numbered and grouped in order of their discovery: groups IB, IIA, IIC, IID, IIE, IIF, III, V, X and XII (*Schaloske & Dennis, 2006*; *Six & Dennis, 2000*). Through their lipase activity, these enzymes trigger various cell-signalling events to regulate cellular functions, directly kill bacteria, or modulate inflammatory responses. In addition, some sPLA2's are high affinity ligands for cellular receptors (*Boyanovsky & Webb, 2009*). Research performed on sPLA2's from snake and

insect venoms have shown that binding targets can include N-type and M-type receptors that mediate myotoxic effects (*Lambeau & Lazdunski, 1999*; *Lambeau et al., 1990*). From the current study, we deduced that COTS sPLA2 proteins exhibited all the key characteristics of sPLA2 (including conserved cysteines, histidine active site, and PLA2 conserved domain) and with relatively high expression in the spines. Furthermore, phylogenetic tree analysis supported clustering of echinoderm sPLA2 with a cnidarian sPLA2 (*Nematostella vectensis*) (*Six & Dennis, 2000*), with closest evolutionary relatedness to I/IX sPLA2. This suggests that COTS sPLA2 might share similar activity to sPLA2 in group I/IX where toxin activities of some members have been reported (*Kelley, Crowl & Dennis, 1992*; *McIntosh et al., 1995*).

A single PI16 (oki.208.17) was found in the COTS spine preparations with almost exclusive gene expression in the spine. Based on its protein sequence and conserved domains, it belonged to the cysteine-rich secretory proteins, antigen 5, and pathogenesis-related 1(CAP) superfamily, a group of proteins that are found across different kingdoms of organisms where their functions are diverse (*Abraham & Chandler, 2017*; *Gaikwad et al., 2020*). Eleven subfamilies have been classified for proteins in the CAP superfamily (*Smith et al., 2017*). These include venom allergen proteins (Ag) and PI16 proteins(previously known as CRISP9), to which COTS PI16 protein showed highest structural similarity. Although COTS PI16 contained a venom allergen 5-like domain, which is a conserved domain found in insect venom proteins (specifically in the Vespidae family) (*Lu et al., 1993*), and a CRISP domain (conserved in CRISP subfamily members *e.g.*, venom-type CRISPs in snakes *Tadokoro et al., 2020*), we found that the COTS PI16 shared higher overall amino acid similarity and closer evolutionary relationship to vertebrate PI16/PI16-like proteins. With the addition of PI16-like proteins we identified in other echinoderm species (within NCBI Nr database), phylogenetic analysis supported the relatedness of echinoderm PI16's with vertebrate PI16/PI16-lke proteins. There have been a number of reports on the functions of PI16 in vertebrates, including its involvement in cardiac function (*Regn et al., 2016*), immune regulation (*Hope et al., 2019*), inflammation (*Nicholson et al., 2012*), the inhibition of matrix metalloproteinase (*Hazell et al., 2016*), and the establishment of neuropathic pain (*Singhmar et al., 2020*). In invertebrates, PI16-like proteins have been identified in the venom gland of the cone snail (*Conus consors*) (*Leonardi et al., 2012*) and the nematocyst of a parasitic cnidarian *Ceratonova shasta*, where PI16 has been proposed to be involved in host immune silencing during replication of the parasite (*Americus et al., 2021*). While the functions of PI16-like proteins in echinoderms have never been reported, the presence of PI16 protein in the COTS spine, as well as its gene expression pattern mirroring that of other COTS toxin proteins, suggests this protein might either be a toxin itself or associated with toxin activity in the COTS spine.

Another group of secreted proteins identified from COTS aboral spine preparations, yet not well-characterized as toxins, included 3 vitellogenins (Vtg), a pentraxin fusion protein-like, a DMBT1 and a keilin/chordin-like protein, which all had relatively low levels of expression in the spine. This raised a question as to whether these proteins might be produced elsewhere prior to transportation to the spines, where they are stored and eventually released. Vtg's are typically recognised as egg-yolk precursors found in egg-laying animals (*Byrne, Gruber & Ab, 1989*), including echinoderms (*Alqaisi et al., 2016*; *Nishimiya*

*et al., 2019*). In most cases, Vtg's are synthesized from tissues distant to the ovary, then transferred *via* the internal circulatory system to oocytes, where it is stored (*Byrne, Gruber & Ab, 1989*). For asteroids, *Vtg* transcripts are synthesized in the follicle cells and pyloric caeca, while the protein products were stored in the eggs (*Alqaisi et al., 2016*). Hence, it is possible that Vtg products detected in the COTS spines might be produced from other tissues before being distributed to the spines. Other functions of Vtg, besides yolk production have been reported, including immune defence and anti-bacterial activity (*Shi, Zhang & Pang, 2006*; *Singh et al., 2013*; *Sun & Zhang, 2015*).

In other phyla, *Dmbt1* is a gene that encodes alternatively spliced glycoproteins associated with membrane or products of epithelial secretions. These glycoproteins have been individually identified in both terrestrial and aquatic animal species (named DMBT1, salivary agglutinin (SAG), crp-ductin, gp-340, ebnerin, vomeroglandin, hensin, and muclin), where they have a variety of innate immune defence functions (*Shiroma et al., 1994*). The COTS genome contains 56 genes that encode proteins annotated as DMBT1-like (*Hall et al., 2017*), these contain a variable number (between 1 and 13) of SR domains, and may also have a signal peptide, transmembrane domains, DDE_Tnp_1 domain, LDLa domain, and/or other domains (Table S3).

## CONCLUSIONS

In this study, we report the presence of toxin protein products derived from the COTS aboral spine, followed by an ultrastructure investigation and proteomic analysis. Spine toxins were broadly categorised into having a molecular weight >10 kDa. A suite of secretory proteins released from the spine have been identified and annotation supports their role in chemical defence. These include known well known COTS toxins, the PLA2 and plancitoxins, as well as those that should be further assessed for toxicity including EPDRs, a PI16-like protein and an uncharacterized protein. The identification of other proteins, derived from spine is of interest as their potential role in defence, or possibly as an alarm response cue used for conspecific signaling.

## ACKNOWLEDGEMENTS

We acknowledge the technical assistance of staff at the UniSC aquarium for housing and husbandry of animals.

### Funding

Adam Hillberg was supported by the Okinawa Institute of Science and Techonology, Graduate University (Japan) POC Program. The funders had no role in study design, data collection and analysis, decision to publish, or preparation of the manuscript.

### Grant Disclosures

The following grant information was disclosed by the authors:
Okinawa Institute of Science and Techonology, Graduate University (Japan) POC Program.

## Competing Interests

Min Zhao is an Academic Editor for PeerJ.

## Author Contributions

- Adam K. Hillberg conceived and designed the experiments, performed the experiments, analyzed the data, prepared figures and/or tables, authored or reviewed drafts of the article, and approved the final draft.
- Meaghan K. Smith conceived and designed the experiments, performed the experiments, analyzed the data, prepared figures and/or tables, authored or reviewed drafts of the article, and approved the final draft.
- Blake S. Lausen conceived and designed the experiments, performed the experiments, analyzed the data, prepared figures and/or tables, authored or reviewed drafts of the article, and approved the final draft.
- Saowaros Suwansa-ard conceived and designed the experiments, performed the experiments, analyzed the data, authored or reviewed drafts of the article, and approved the final draft.
- Ryan Johnston conceived and designed the experiments, performed the experiments, analyzed the data, prepared figures and/or tables, authored or reviewed drafts of the article, and approved the final draft.
- Shahida A. Mitu conceived and designed the experiments, performed the experiments, analyzed the data, authored or reviewed drafts of the article, and approved the final draft.
- Leah E. MacDonald conceived and designed the experiments, performed the experiments, analyzed the data, authored or reviewed drafts of the article, and approved the final draft.
- Min Zhao performed the experiments, analyzed the data, authored or reviewed drafts of the article, and approved the final draft.
- Cherie A. Motti analyzed the data, authored or reviewed drafts of the article, and approved the final draft.
- Tianfang Wang conceived and designed the experiments, analyzed the data, authored or reviewed drafts of the article, and approved the final draft.
- Abigail Elizur analyzed the data, authored or reviewed drafts of the article, and approved the final draft.
- Keisuke Nakashima analyzed the data, authored or reviewed drafts of the article, and approved the final draft.
- Noriyuki Satoh analyzed the data, authored or reviewed drafts of the article, and approved the final draft.
- Scott F. Cummins conceived and designed the experiments, analyzed the data, prepared figures and/or tables, authored or reviewed drafts of the article, and approved the final draft.

## DNA Deposition

The following information was supplied regarding the deposition of DNA sequences:

Raw sequence data are available at the NCBI under Genbank: PRJNA901199.

## Data Availability

The sequence data is available at NCBI: PRJNA901199.

Toxicity assay data and raw gene expression measurements are available in the Supplementary Files.

## Supplemental Information

Supplemental information for this article can be found online at http://dx.doi.org/10.7717/peerj.15689#supplemental-information.

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
