# Peer review of "Crown-of-thorns starfish spines secrete defence proteins"

_PeerJ, doi:10.7717/peerj.15689_

## Round 0.1 · original submission · Minor Revisions

Dear authors,

Two reviewers were positive concerning your paper. Please make the changes proposed by the reviewers and resubmit your article.

Reviewer 1 ·

Basic reporting

no comment

Experimental design

no comment

Validity of the findings

no comment

Additional comments

In this manuscript, the authors report the observation of the ultrastructure of the spines secreted toxins in the Crown-of-Thorns Starûsh (COTS). Toxins derived from COTS have lethal effect on brine shrimp. The components of toxins from COTS was revealed by venomics, and their expression patterns were profiled. The results provide insight into the evolution of toxins of COTS and are helpful for discovering biomolecules from them with potential applications in medicine.

Major issue:
According to the current description in the materials and methods section, I can’t get how the proteins secreted by spines were collected for identification using the LC-MS/MS. If the proteins are not secreted by spines and they were collected by homogenizing the spines, I want to see how the authors to make sure that the identified components are the true toxins. Also, the authors should provide more information to let us know how to identify the toxins by venomics, via the proteomic and transcriptomic data. Most of the information about the proteomic and transcriptomic methods are not related to how to reveal the toxins from the data. If possible, qPCR or other analyses are required to validate the toxins deciphered via venomics.

Minor issues:
1. There are typos and grammatical errors in the text. Here, I jus pick some of them.
Line 47, line 54, e.g.--e.g.,
Line 59, provide--provides
Line 72, their properties clarified-- their properties have been clarified
Line 83, 2--two
Line 89, 3--three
2. The format of some references is not fit to the requirement of the journal.

Reviewer 2 ·

Basic reporting

The paper is well written and provides very interesting insights into the gene expression in different compartments on Crown of Thorns starfish and protein expression within the spines. The reference list needs checking citation 26 has not been inserted into the paper and some of the journal titles need capitalisation (eg.ref 9) and/or abbreviation ( eg. Ref 7, 9, 12 14 ).

The figures and raw data presented was mostly easy to follow and relevant to the aims of the paper.

Experimental design

The design was mostly clear however I had a few questions.
1. Line 83 what is a protein skimmed isolated saltwater system?
2. The biological and technical replications needs clarifying. Line 88 writes about 10 spines from 3 different adult COTS and suggests all samples are pooled and then fractionated. In Table s1, it look like there are three biological replicates , I guess these match fractionation from the three different adults, but when these are then added to the toxicity assay, n =5 so is this a pool of the three biological replicates added to five different well containing artemia? Please clarify. I note later that proteomics was performed on three separate samples.
3. Throughout the methods and results it needs to be clarified that the conc in well is equivalent to protein concentration estimation ug.ml as this was not clear it could have been wet extract weight. Also if protein measurements were done on individual samples how does this relate to the 5 replicates in the toxicity assay, are they biological replicates pooled and protein concentration remeasured or ? actually now that I think about it absorbance 280 nm on a Nanodrop is not a very good measure of protein concentration (ug/ml), normally I would expect a BCA or a Bradford estimation to be done. It is good that standardisation is used but perhaps this should just be called OD280nm or acknowledging that this is very rough estimation of protein concentration.
4. Line 115 suggest that 5 replicates per dose per extract, so that would be 5 x 3 x 3 but this does not seem to match Table S 1. I can see 5 x each fraction and dose but not from three different individuals so perhaps this is pooled?
5. Line 118 insert word “added to the well”
6. Line 155 it would be good to explain in more detail the relevance of relaxed and stressed individuals, how is agitation stress relevant to their aim of finding toxins being releases by the COTS – are these expected to be defence toxins?.
7. Spines were acidified with TFA and underwent trypsin solution, given that spines are quite solid, I was surprised that no mechanical force, grinding or crushing was applied to the spine. This may have been deliberate as to only collect secreted proteins, but some clarification of this would be good.

Validity of the findings

As outlined in point 7 above it is unclear if the aboral spine proteins detected are only those secreted in to the spaces in the spines or whether some of the proteins that form the spine scaffold would/should be detected. From looking at the list there are no collagens or keratin type proteins so probably only the proteins present in the spaces in the spine have been collected/detected, this may explain the low number of protein sequences detected 157.
Figure 1 is 1 A- a photo or a digital rendering of a COTS or ?
In Table 1 LC50 and in Figure 1 B needs to be clear that ug?ml is protein concentration in the legends.
Figure 3 it is a little unclear how the oki mRNA transcripts displayed are chosen to be represented in this figure. There are 16 genes displayed that form two clades. The legend states it is the gene expression of the proteins identified in the proteome but 157 proteins were identified. Also further data representing the three toxins was presented in Fig 4, 5 and 6, so It is unclear why some of the Oki transcript expression was presented in one figure over the other (occasionally data was presented in both). Given there was only 157 proteins identified why not show the gene expression of them all?
Figure 4 and 5 legend have a line “Red asterisks denote thosed identified in this study by aboral spine protein analysis” but I cannot see any red asterisks on the pdf on screen in colour.
Overall the discussion and conclusions were easy to follow and summarised the highlights of the work.

Additional comments

Overall a very interesting piece of work it was particularly pleasing to see the Supplementary tables were supplied that were easy to follow and also provided the amino acid sequences of the spine proteins that were identified.

---

## Round 0.2 · accepted · Accept

I think the article is publishable in its current condition.